# First Detection of Severe Acute Respiratory Syndrome Coronavirus 2 on the Surfaces of Tourist-Recreational Facilities in Italy

**DOI:** 10.3390/ijerph18063252

**Published:** 2021-03-21

**Authors:** Maria Teresa Montagna, Osvalda De Giglio, Carla Calia, Chrysovalentinos Pousis, Francesca Apollonio, Carmen Campanale, Giusy Diella, Marco Lopuzzo, Angelo Marzella, Francesco Triggiano, Vincenzo Marcotrigiano, Domenico Pio Sorrenti, Giovanni Trifone Sorrenti, Pantaleo Magarelli, Giuseppina Caggiano

**Affiliations:** 1Department of Biomedical Science and Human Oncology—Hygiene Section, University of Bari Aldo Moro, Piazza G. Cesare 11, 70124 Bari, Italy; mariateresa.montagna@uniba.it (M.T.M.); osvalda.degiglio@uniba.it (O.D.G.); carla.calia@virgilio.it (C.C.); vpousis@gmail.com (C.P.); francesca.apo@libero.it (F.A.); campanale.carmen@libero.it (C.C.); giusy.diella@uniba.it (G.D.); marcolopuzzo@gmail.com (M.L.); marzella.angelo@libero.it (A.M.); francesco.triggiano@uniba.it (F.T.); 2Department of Prevention, Food Hygiene and Nutrition Service, Local Health Unit BT, Barletta-Andria-Trani, 76125 Trani, Italy; vincenzo.marcotrigiano@aslbat.it (V.M.); d.sorrenti2@studenti.uniba.it (D.P.S.); giovanni.sorrenti@aslbat.it (G.T.S.); pantaleo.magarelli@aslbat.it (P.M.)

**Keywords:** COVID-19, SARS-CoV-2, tourist-recreational facilities, respiratory infections, environment, surface

## Abstract

A Coronavirus disease (COVID-19), caused by a new virus called severe acute respiratory syndrome coronavirus 2 (SARS-CoV-2), spreads via direct contact through droplets produced by infected individuals. The transmission of this virus can also occur via indirect contact if objects and surfaces are contaminated by secretions from individuals with COVID-19 or asymptomatic carriers. Environmental contamination with SARS-CoV-2 is high in hospital settings; on the contrary, surface contamination in non-healthcare settings is still poorly studied. In this study, the presence of SARS-CoV-2 on the surfaces of 20 tourist-recreational facilities was investigated by performing a total of 100 swabs on surfaces, including refrigerator handles, handrails, counters, tables, and bathroom access doors. Six (6%) swabs from four (20%) tourist-recreational facilities tested positive for SARS-CoV-2; the surfaces that were involved were toilet door handles, refrigerator handles, handrails, and bar counters. This study highlights that SARS-CoV-2 is also present in non-healthcare environments; therefore, in order to limit this worrying pandemic, compliance with behavioral rules and the adoption of preventive and protective measures are of fundamental importance not only in healthcare or work environments but also in life environments.

## 1. Introduction

In January 2020, the World Health Organization (WHO) recognized a disease (coronavirus disease (COVID-19)) caused by a new virus called severe acute respiratory syndrome coronavirus 2 (SARS-CoV-2) belonging to the Coronaviridae, a large family of viruses that can cause illness ranging from the common cold to more severe diseases. In February 2020, SARS-CoV-2 was considered the third greatest pathogenic human coronavirus that emerged in the last two decades [1].

In Italy, the first cases of COVID-19 were recognized in February 2020. On March 11, following the rapid spread of SARS-CoV-2 infection, the WHO decreed a pandemic status [2]. On the same day, Italy extended the measures envisaged by the previous decrees to the entire national territory [3], declaring a state of lockdown aimed to limit the movement of individuals only to absolute necessity. So far, this restriction increased the risk of adopting sedentary behaviors that reduced people’s physical activity [4].

After a brief reduction in its spread during the summer, SARS-CoV-2 continued to rapidly spread, causing an increase in the number of cases, with 2,611,659 total cases confirmed and 90,618 dead patients registered until February 5, 2021 [5]. 

The COVID-19 pandemic has completely threatened the health system. In fact, hospitals had to deal with an extraordinarily high frequency of patients in intensive care units and an abnormal request for mechanical ventilators and beds intended for infectious disease units. Thus, they rapidly adapted the available spaces to quickly create new beds [6,7].

The transmission of SARS-CoV-2 occurs through droplets produced by infected individuals via breathing, speaking, coughing, and sneezing. Owing to their size, the droplets travel in the air for short distances, generally less than 1 m, and can directly reach susceptible subjects who are in the immediate vicinity [8]. SARS-CoV-2 can also be transmitted via indirect contact if objects and surfaces are contaminated by secretions from individuals with COVID-19 or asymptomatic carriers, especially in the hospital environment [9,10,11,12]. Some authors have shown that this virus survives for hours or days on surfaces, depending on the type of material, humidity, and temperature [13]. The SARS-CoV-2 virus has been found to survive on plastic, stainless steel, and other surfaces from several hours to several days [12,13]. Patients’ and healthcare workers’ hands, even if equipped with gloves, can be a vehicle of contagion spreading after touching surfaces [14].

On the contrary, surface contamination in non-healthcare settings is still poorly studied. In fact, to date, community environments have been studied with reference to episodes of infection, while as far as we know no recreational facilities have been targeted for investigation, especially after sanitation [15]. These premises lead us to reflect about investigating community environments frequented by many people.

The aim of our study was to investigate the presence of SARS-CoV-2 on the surfaces of tourist-recreational facilities, taking into account the peculiarity of these settings, in which many individuals pass through, stop, and—in order to consume foods and drinks—may also temporarily not wear protective equipment.

## 2. Materials and Methods

Environmental surface specimens were collected from September to October 2020 in 20 tourist-recreational facilities located in a coastal geographic context of the Apulia region, which was randomly selected among the most frequented locations in the summer season. In total, 100 surface samples were examined: 22 swabs obtained from five bars and pastry shops, 23 swabs from five beach resorts with an annexed coffee cart, 33 swabs from five accommodation facilities with an annexed restaurant, 18 swabs from four restaurants/banquet halls, and 4 swabs from one seafront coffee cart.

The sampling was conducted using sterile swabs inserted into a plastic tube (Easy Surface Checking [ESC] swab—Neutralizing Rinse Solution [NRS]; Liofilchem Srl, Roseto degli Abruzzi (Teramo), Italy) containing 10 mL transport medium, selecting points considered to be of greater high-touch surface representativeness, such as bar counters, food consumption tables, refrigerator handles, handrails, elevator pushbuttons, and toilet access door handles. For the sampling, moistened swabs were rotated over a standard sample area (10 × 10 cm) of the selected surface and transported to the laboratory in a special isothermal refrigerator at +4 °C, where they were immediately stored at −80 °C and tested after three days at the latest. Before processing, all samples were thawed at +22 °C, vigorously vortexed for 20 s, and transferred under sterile conditions to a new 15-mL capacity tube.

In according to data literature [16], SARS-CoV-2 detection was conducted via molecular investigation using real-time reverse-transcription PCR (RT-PCR).

RNA extraction: Nucleic acids were extracted from 5 mL of the NRS medium using the NucliSENS miniMAG semiautomatic extraction system with magnetic silica, in accordance with the manufacturer’s instructions (bioMerieux, Marcy-l’Etoile, Lione- France). The RNAs were resuspended in 100 μL of elution buffer and stored in aliquots at −20 °C.

Real-time RT-PCR: A 25-μL reaction contained 5 μL of RNA; 12.5 μL of 2× reaction buffer provided with the AgPath-ID™ One-Step RT-PCR Reagents (Applied Biosystems™, ThermoFisher, MA, USA); 1 μL of 25X RT-PCR Enzyme Mix; 1 μL of forward primer (12.5 μM); 1 μL of reverse primer (22.5 μM); 1 mL of probe (6.25 μM); 1.83 μL of nuclease-free water (not DEPC-treated); and 1.67 μL of detection enhancer for real-time PCR (Applied Biosystems™, ThermoFisher, MA, USA). The following primer and probe sequences were used: CoV-2-F/ACA TGG CTT TGA GTT GAC ATC T; CoV-2-R/AGC AGT GGA AAA GCAT GTG G; CoV-2-P/FAM-CAT AGA CAA CAG GTG CGC TC-MGBEQ [16]. 

The region amplified by these primers is the ORF-1ab gene (nsp14). The qRTPCR experiments were conducted in triplicate using the CFX96 Touch Deep Well Real-Time PCR System (Applied Biosystem™, ThermoFisher, MA, USA). The thermal cycling conditions included an initial reverse transcription step at 50 °C for 30 min, inactivation of the RT step at +95 °C for 10 min, and 45 cycles of amplification at +95 °C for 15 s and +60 °C for 45 s. The cycle threshold values for the RTPCR were used as indicators of the copy number of SARS-CoV-2 RNA in the samples, with lower cycle threshold values corresponding to higher viral copy numbers. A cycle threshold value of less than 40 was interpreted as positive for SARS-CoV-2 RNA.

## 3. Results

Of the 20 tourist-recreational facilities, four (20%) yielded positive results for SARS-CoV-2: beach resorts with annexed coffee cart (2/23 swabs, 8.7%), accommodation facilities with annexed restaurant (2/33 swabs, 6.0%), restaurants/banquet halls (1/18 swabs, 5.5%), and bars and pastry shops (1/22 swabs, 4.5%) (Table 1).

The Cycle threshold (Ct) values of the positive samples were 37.86 and 38.06, respectively, from refrigerator handles and handrails in beach resorts with a coffee cart.

For toilet access door handles and refrigerator handles from accommodation facilities with a restaurant, the Ct values were 38.05 and 39.15, respectively.

Finally, for toilet access door handles from restaurants/banquet halls the Ct value was 38.70, and in bar counters from bars and pastry shops the Ct value was 38.59.

## 4. Discussion

Some authors suggest that environmental contamination with SARS-CoV-2 is high in hospital settings where patients with COVID-19 are hospitalized [11,12,17,18]. Contamination of surfaces by asymptomatic individuals has also been demonstrated [10]; however, it is not yet clear whether asymptomatic individuals as well as patients with COVID-19 can contaminate the environment [11].

To our knowledge, this study is the first investigation on the presence of SARS-CoV-2 on the surfaces of tourist-recreational facilities. During the summer, many individuals of all ages attend these areas, including asymptomatic individuals. The first sampling was conducted when the Apulia region recorded 1915 positive cases for SARS-CoV-2 [5]. The daily increase in the number of new cases at that time averaged approximately 100 cases per day. Our last sampling was conducted 20 days after the first, when the number of both symptomatic and asymptomatic patients with COVID-19 had reached 3133. Our results show an increasing trend of SARS-CoV-2-positive surfaces, parallel to the increase in the number of patients with positive results.

We wish to highlight that three (refrigerator handles and toilet access door handles) of the six positive swabs were performed on surfaces accessible to food handling staff only. The transmission of the virus through food contamination has not been documented to date; however, our data suggest that surfaces can be contaminated and that hand hygiene and environmental disinfection play an essential role in preventing infectious diseases, particularly airborne ones. Although there are no data on the transmissibility of coronaviruses from contaminated surfaces to hands [19], as well as of the influenza A virus [20], SARS-CoV-2 can survive for several hours to several days on different surfaces, and it is viable for up to 72 h on plastic, 48 h on stainless steel, and 24 h on cardboard [12,13,19]. On the contrary, copper surfaces tend to kill the virus in about 4 h. 

Some authors evaluated the stability of SARS-CoV-2 at different temperatures, showing that the virus is highly stable at +4 °C but that it is sensitive to heat. Further studies confirmed that SARS-CoV-2 lost its infectivity after 90-, 60-, and 30-min exposures to temperatures of +56 °C, +67 °C, and +75 °C, respectively [21].

In particular, the virus appears to be more stable on smooth surfaces and extremely stable over a wide range of pH values (pH 3–10) at room temperature (+20 °C) [22,23]. To date, the guidelines of the European Centre for Prevention and Disease, the United States Centers for Disease and Control [24,25], and WHO designate that cleaning with water and normal neutral detergents associated with disinfectants is sufficient for the decontamination of surfaces. In general, it has been shown that alcohol-based disinfectants (e.g., ethanol, propan-2-ol, and propan-1-ol) or sodium hypochlorite can significantly reduce the viral load of SARS-CoV-2.

In this study, the surfaces that tested positive for SARS-CoV-2 were steel and wood. Although these materials can promote the survival of the virus from one to four days, a more accurate application of sanitation processes would have reduced its circulation in the environment. Furthermore, it is necessary to deeply assess the update of Hazard Analysis and Critical Control Points plans (HACCP) during official controls, in order to be sure of their adequacy in this health emergency context in which proper disinfection and sanitization procedures must be put in place by Food Business Operators [26,27,28]. 

## 5. Conclusions

This study highlights the high prevalence of SARS-CoV-2, even in non-healthcare settings. Although there is no comprehensive evidence to scientifically demonstrate the survival and transmission of the virus through contact with contaminated surfaces, we cannot exclude such a possibility. Although the attention on the COVID-19 pandemic is rightly mainly aimed at health settings, the environments investigated in this study must continue to be closely monitored because viral circulation in social contexts plays an essential role in the spreading of new infections. New and more accurate studies are needed to combat this ‘monster’.

## Figures and Tables

**Table 1 ijerph-18-03252-t001:** Swabs from tourist-recreational facilities that yielded positive results for severe acute respiratory syndrome coronavirus 2.

Examined Facilities (No.) - Positive Surface (No.)	Swabs No. of Positive Results/Total No. (%)
**Beach resorts with coffee cart (5)** - refrigerator handles (1) - handrails (1)	2/23 (8.7)
**Accommodation facilities with restaurant (5)** - toilet access door handles (1) - refrigerator handles (1)	2/33 (6.0)
**Restaurants/banquet halls (5)** - toilet access door handles (1)	1/18 (5.5)
**Bars and pastry shops (4)** - bar counters (1)	1/22 (4.5)
**Coffee cart seafront (1)**	0/4 (0)
**Total**	6/100 (6)

## Data Availability

Please refer to suggested Data Availability Statements in section “MDPI Research Data Policies” at https://www.mdpi.com/ethics (accessed on 5 February 2021).

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
