# Peer review of "First Detection of Severe Acute Respiratory Syndrome Coronavirus 2 on the Surfaces of Tourist-Recreational Facilities in Italy"

_ijerph, 2021, doi:10.3390/ijerph18063252_

Round 1

Reviewer 1 Report

In this manuscript,  Montagna et al. reported that tourist-recreational facilities can be contaminated by SARS-CoV-2 in Italy and suggest that hand hygiene and environmental disinfection play an essential role in preventing infectious diseases pandemic in public. The idea is clear. The conclusions can be supported by the data and the manuscript is well organized and written. However, the reviewer doesn't recommend it for publication in this journal due to the lack of significance and innovation.

  1. In the Introduction, the significance of surface contamination in non-healthcare settings is not fully illustrated.
  2. There is no significant difference between this study with other works about testing the SARS-CoV-2 stability on various material surfaces.

Author Response

In the Introduction, the significance of surface contamination in non-healthcare settings is not fully illustrated.

We thank the Reviewer for comments which allow us to point out some aspects of the study.

The changes are reported in the introduction paragraph (lines 68-71)

There is no significant difference between this study with other works about testing the SARS-CoV-2 stability on various material surfaces.

At our knowledge this is the first study that focuses on viral diffusion on the surfaces tourist-recreational facilities, while other Authors tested the SARS-CoV2 stability on several surfaces made of different materials but the latter doesn’t represent the aim of our study.

Reviewer 2 Report

In this study authors  describe the extent of SARS-CoV-2 environmental contamination from September to October 2020 in 73  tourist-recreational facilities. Many studies have highlighted virus presence on surfaces confirming  that they represent a potential vehicle of transmission. Moreover, futher studies identified airborne as the primary route of transmission of SARS-CoV-2. 

The only novelty of the study is having analyzed surfaces in tourist facilities. In order to be original, the study would need to analyze more samples over the same period. Alternatively, one could compare the September/October viral load with that of another time of year. Moreover, the positivity of 6/100 samples is very low. Considering that for the RTPCR  the cycle threshold value of less than 40 is too high, probably the positivity would drop to zero setting the cycle threshold value of  35.

Author Response

The only novelty of the study is having analyzed surfaces in tourist facilities. In order to be original, the study would need to analyze more samples over the same period. Alternatively, one could compare the September/October viral load with that of another time of year. Moreover, the positivity of 6/100 samples is very low. Considering that for the RT-PCR  the cycle threshold value of less than 40 is too high, probably the positivity would drop to zero setting the cycle threshold value of  35.

We thank the Reviewer for the comments, we agree on the low number of samples, but, at our knowledge, this is the first study that analyses the tourist facilities surfaces.Moreover, these structures were frequented by a large number of customers, then were sanitized and later closed, but in spite of that some samples were positive. Our project is to improve our study starting from the next summer season, because during other part of the year these structures were not fully open due to restrictions adopted from our Government to restrict the pandemic.

Reviewer 3 Report

The paper by Montagna et al described the detection of SARS-CoV-2 in recreational facility. Since the manuscript seem to be attractive to a wide readership, it must be reviewed. 

Introduction:

L49: substitution of 'to date' with the precise date in which the number of cases and deaths was registered.

L52: font formatting  of 'mechanical ventilators'

L62 (and 141): dash between SARS and CoV

Materials and methods

L80-82: the authors used ESC swab- neutralizing rinse solution for swabbing surfaces. This method is useful for bacterial detection, but it is not standardized for virus detection. The molecules contained in the medium are necessary to neutralize disinfectants, but they are not tested (to my knowledge) for viral transport. A proof is needed to demonstrate that also RNA viruses can be detected with this medium at the same rate of the classic viral transport media (by literature or by a comparative test in your lab). The limit here is the possibility to underestimate the number of positive samples due to an incorrect transport medium. 

L101-102: which is the SARS-CoV-2 region amplified by these primers?

Results

If you don't have an absolute quantification with standard curve for this qRT-PCR, it could be helpful adding the Ct values of the positive sample. furthermore, it could be interesting to evaluate if there was any difference in viral load according to the different surface samples.

discussion

L145: capital letter

In conclusion, the results obtained in this paper are interesting, but the risk of an underestimation of the rate of SARS-CoV-2 presence in surfaces is relevant. To overcome this possible problem the authors could find in the literature other studies that use the same surface swabbing for RNA virus detection. As an alternative they could demonstrate by a comparative test the rate of detection of viruses between the method used in this study and a more appropriate viral transport medium. It is not necessary to use SARS-CoV-2, but any low risk RNA virus.

Round 2

Reviewer 1 Report

In this manuscript, Montagna et al. reported the first study that focuses on viral diffusion on the surfaces of tourist-recreational facilities. Six swabs from 100 samples were tested positive for SARS-CoV-2, demonstrated the virus spread in the recreational facilities even after sanitation, and highlighted that hand hygiene and environmental disinfection play an essential role in preventing the pandemic. The idea is clear. The conclusions can be supported by the data and the manuscript is well organized and written. The reviewer recommends it for publication in this journal.

Author Response

We thank the Reviewer for the kind comments about our manuscript

Reviewer 2 Report

The authors have responded to some of the queries satisfactorily but one question still remain listed below, please clarify:

The positivity of 6/100 samples is very low. Considering that for the RT-PCR the cycle threshold value of less than 40 is too high, probably the positivity would drop to zero setting the cycle threshold value of 35.

Author Response

We thanks the Reviewer for kind comment. Yes, it is true that if we will
be setting the cycle threshold value of 35 the positivity would drop to
zero. we would like specify that decided to setting the threshold value at 40 proceeded according to other authors which established both Thermal cycling conditions and ct value in order to evaluate the presence of SARS CoV 2 in the surfaces as well as in environmental samples. (We specified it at paragraph material and methods - line 94, highlighted in yellow)

Here other articles where the Thermal cycling conditions are the same with
those used by us and they have considerate as positive all the samples with
qRTPCR Ct value ≤40

Katia Razzini et al 2020 : SARS-CoV-2 RNA detection in the air and on
surfaces in the COVID-19 ward of a hospital in Milan, Italy

Po Ying Chia et al 2020: Detection of air and surface contamination by
SARS-CoV-2 in hospital rooms of infected patients

Giuseppina La Rosa et al. 2020: First detection of SARS-CoV-2 in untreated
wastewaters in Italy.